# The Glucose–Succinate Pathway: A Crucial Anaerobic Metabolic Pathway in the Scallop *Chlamys farreri* Experiencing Heat Stress

**DOI:** 10.3390/ijms25094741

**Published:** 2024-04-26

**Authors:** Lijingjing Bao, Zhi Liu, Mingyi Sui, Zujing Yang, Haoran Wang, Xiaofei Chen, Yue Xu, Zehua Niu, Na Liu, Qiang Xing, Zhenmin Bao, Xiaoting Huang

**Affiliations:** 1MOE Key Laboratory of Marine Genetics and Breeding, College of Marine Life Sciences/Academy of Future Ocean, Ocean University of China, Qingdao 266100, China; 2Laboratory for Marine Fisheries Science and Food Production Processes, Qingdao National Laboratory for Marine Science and Technology, Qingdao 266237, China; 3Laboratory of Tropical Marine Germplasm Resources and Breeding Engineering, Sanya Oceanographic Institution, Ocean University of China, Sanya 572000, China

**Keywords:** heat stress, anaerobic metabolic, *Chlamys farreri*, phosphoenolpyruvate carboxykinase, glucose–succinate pathway

## Abstract

Recently, the increase in marine temperatures has become an important global marine environmental issue. The ability of energy supply in marine animals plays a crucial role in avoiding the stress of elevated temperatures. The investigation into anaerobic metabolism, an essential mechanism for regulating energy provision under heat stress, is limited in mollusks. In this study, key enzymes of four anaerobic metabolic pathways were identified in the genome of scallop *Chlamys farreri*, respectively including five opine dehydrogenases (CfOpDHs), two aspartate aminotransferases (CfASTs) divided into cytoplasmic (CfAST1) and mitochondrial subtype (CfAST2), and two phosphoenolpyruvate carboxykinases (CfPEPCKs) divided into a primitive type (CfPEPCK2) and a cytoplasmic subtype (CfPEPCK1). It was surprising that lactate dehydrogenase (LDH), a key enzyme in the anaerobic metabolism of the glucose–lactate pathway in vertebrates, was absent in the genome of scallops. Phylogenetic analysis verified that CfOpDHs clustered according to the phylogenetic relationships of the organisms rather than substrate specificity. Furthermore, *CfOpDHs*, *CfASTs*, and *CfPEPCKs* displayed distinct expression patterns throughout the developmental process and showed a prominent expression in muscle, foot, kidney, male gonad, and ganglia tissues. Notably, *CfASTs* displayed the highest level of expression among these genes during the developmental process and in adult tissues. Under heat stress, the expression of *CfASTs* exhibited a general downregulation trend in the six tissues examined. The expression of *CfOpDHs* also displayed a downregulation trend in most tissues, except *CfOpDH1/3* in striated muscle showing significant up-regulation at some time points. Remarkably, *CfPEPCK1* was significantly upregulated in all six tested tissues at almost all time points. Therefore, we speculated that the glucose–succinate pathway, catalyzed by *CfPEPCK1*, serves as the primary anaerobic metabolic pathway in mollusks experiencing heat stress, with *CfOpDH3* catalyzing the glucose–opine pathway in striated muscle as supplementary. Additionally, the high and stable expression level of *CfASTs* is crucial for the maintenance of the essential functions of aspartate aminotransferase (AST). This study provides a comprehensive and systematic analysis of the key enzymes involved in anaerobic metabolism pathways, which holds significant importance in understanding the mechanism of energy supply in mollusks.

## 1. Introduction

In the 1970s, China’s mariculture industry began cultivating seawater animals such as scallops [1]. Scallops have delicious meat and rich nutritional value [2,3]. Scallops harvested in China account for more than 80% of the total global annual production. According to the statistics of the China Fishery Statistical Yearbook 2022, the aquaculture output of mollusks in China reached 1.83 million tons in 2021, an increase of 4.81 percent year-on-year [4]. Currently, the predominant source of scallop production is aquaculture, with a lesser contribution from traditional fishing methods. Additionally, scallops can contribute to environmental sustainability by filtering water and providing a habitat for various organisms [5]. Scallops possess considerable ecological importance, yet alterations in the environment can also influence the productivity and viability of scallop aquaculture.

According to the data released by C3S (Copernicus Climate Change Service) in 2023, the global average sea surface temperature experienced a continual increase, culminating in a historic peak in July. Throughout July, the global mean sea surface temperature exceeded the 1991–2020 average by 0.51 °C. Marine mollusks belong to ectotherms and are particularly sensitive to alterations in their surrounding environment [6]. High temperatures in seawater can hinder the growth of marine mollusks and even precipitate mass mortality events [7]. Marine invertebrates exhibit various adaptive mechanisms in response to heat stress for survival [7,8,9]. Among them, energy metabolism is crucial in enabling marine invertebrates to effectively cope with heat stress. Previous studies have shown that elevated sea temperatures may enhance the metabolic rates and energy expenditure of marine animals while simultaneously reducing the activity of some aerobic energy metabolism pathways, consequently affecting their typical daily activities [10,11]. How to maintain a balance between increased energy requirements and reduced aerobic metabolism under heat stress becomes a crucial concern for marine organisms to ensure survival, with anaerobic metabolism serving as a vital alternative pathway [10,12,13].

The increasing sea temperatures can lead to a decrease in the oxygen solubility in seawater [14,15]. Under hypoxia stress conditions, marine vertebrates, such as fish, enhance glycolysis and utilize the glucose–lactate anaerobic respiration pathway as the main supplementary pathway [16]. Marine invertebrates, such as mollusks, utilize four anaerobic respiration pathways: glucose–lactate, glucose–opine, aspartate–succinate, and glucose–succinate [17]. The key enzyme involved in the glucose–lactate pathway is NAD-dependent L-lactate dehydrogenase (L-LDH; EC 1.1.1.27), which, assisted by NADH, catalyzes the conversion of pyruvate to lactate and NAD^+^ [17,18]. For example, the muscular motion in the *Nereis diversicolor* is supported by the glucose–lactate pathway [19]. The glucose–opine pathway relies on the key enzyme opine dehydrogenase (OpDH; EC 1.5.1.28), which, assisted by NADH, catalyzes the reduction–condensation of pyruvate and amino acids to form amino acid opines [13,17]. Within this pathway, six members of the OpDHs family have been identified based on their substrate specificity: Octopine dehydrogenase (OcDH; EC 1.5.1.11), Strombine dehydrogenase (StDH; EC 1.5.1.22), Tauropine dehydrogenase (TaDH; EC 1.5.1.23), Alanopine dehydrogenase (AlaDH; EC 1.5.1.17), β-Alanopine dehydrogenase (beta-AlaDH; EC 1.5.1.26), and Lysopine dehydrogenase (LyDH) [20,21]. For example, *Cellana toreuma* relies on the glucose–opine pathway for anaerobic metabolism [12]. The key enzymes involved in the aspartate–succinate and glucose–succinate pathways are aspartate aminotransferase (AST; EC 2.6.1.1) and phosphoenolpyruvate carboxykinase (PEPCK; EC 4.1.1.32), respectively. AST catalyzes the conversion of aspartate to oxaloacetate, which ultimately leads to the production of succinate [17,22]. It is worth noting that AST can be further classified into cytosolic isoenzyme AST-C and mitochondrial isoenzyme AST-M based on their subcellular localization [23]. The aspartate–succinate pathway in *Mytilus edulis* sets a good example [24]. PEPCK facilitates the production of oxaloacetate from phosphoenolpyruvate, followed by its transformation into succinate or acetate [25,26,27,28,29]. Within the striated muscle of *Patinopecten yessoensis*, the glucose–succinate pathway serves a crucial function during the middle and late stages (6–12 h) of sustained hypoxia stress that arises from dry storage. Upon careful examination, this pathway releases propionate and acetate through volatilization to prevent material buildup and preserve muscle pH stability, while simultaneously generating ATP and NADH to facilitate glycolysis [30]. Depending on the energy source utilized, PEPCK can be categorized into two classes: ATP-dependent and GTP-dependent. Within the GTP-dependent class, there are cytosolic isoenzyme PEPCK-C and mitochondrial isoenzyme PEPCK-M based on their subcellular localization [31,32,33]. Until now, the primary anaerobic metabolism pathway in mollusks remains incompletely elucidated, with the majority of research on the anaerobic metabolism of marine invertebrates concentrating on enzyme activity and end products of anaerobic metabolism [12,21,23,34,35,36,37,38]. There exists a significant deficiency in systematic and comprehensive research regarding the anaerobic metabolism mechanisms in marine mollusks and their reaction to heat stress, which could be addressed through the rapidly progressed genome and transcriptome datasets.

In this study, based on the whole genome and transcriptome database of *Chlamys farreri*, systematic identification and evolutionary analysis of LDH, OpDH, AST, and PEPCK in *C. farreri* were carried out, and the expression profiles during larval development in adult tissues and under heat stress were investigated. The results would elucidate and characterize the anaerobic metabolism pathway and shed light on its roles in mollusks.

## 2. Results

### 2.1. Gene Identification and Sequence Analysis

Five OpDH genes (named CfOpDH1-5), two PEPCK genes (named CfPEPCK1-2), and two AST genes (named CfAST1-2) were identified from the transcriptome and genome of *C. farreri*, which showed great sequence similarity with that of other species from invertebrate to vertebrate, while no LDH gene was identified. The sequence characteristics of those identified genes were summarized in Table 1, and the detailed protein sequences were listed in Appendix A. The ORFs (open reading frames) of *CfOpDHs* are 1200–1314 bp in length, encoding 339–437 amino acids. The predicted molecular weights of CfPEPCK1 and CfPEPCK2 are 73.47 and 71.66 kDa, with pIs of 7.56 and 8.20, respectively. CfAST1 is composed of nine exons and eight introns, while CfAST2 contains eleven exons and ten introns (Figure 1 and Table 1). The secondary structures of CfOpDHs showed that they respectively consist of 20–26 alpha helixes, 20–34 beta strands, 25–34 coils, and 24–38 turns (Figure 2 and Table 1). The secondary structures of CfPEPCKs showed that they respectively consist of 11–12 alpha helixes, 50–53 beta strands, 70 coils, and 66–78 turns, and those of *CfASTs* were 8–13, 31–32, 38–42, and 42–44 (Figure 3 and Table 1).

Domain analysis showed that there was a NAD(P)-binding domain superfamily (IPR036291) and an opine dehydrogenase domain (IPR003421) in all CfOpDHs (Figure 4). Both CfPEPCKs consist of a PEPCK GTP-utilizing N-terminal domain (IPR035078) and a PEPCK GTP-utilizing C-terminal P-loop domain (IPR035077), with an evolutionarily conserved site (IPR018091, FPSACGKT/SN, marked with red box in Figure 5B) in the center (Figure 4 and Figure 5). There was an AST pyridoxal-phosphate-binding site (IPR004838) in the middle of AST Ⅰ/Ⅱ (IPR004839) in both CfASTs (Figure 4).

Multiple alignments of OpDH revealed that the NAD(P)-binding domain exhibits conserved Rossmann-fold motifs (GXGXXA/G) (Figure 5A). Compared with other species, the GTP-binding site (marked with orange box), substrate-binding site (marked with green box), and metal-binding site (marked with line box) of CfPEPCK were also identified and similar to others (Figure 5B). The alignments of AST showed some shared sites: the pyridoxal-phosphate-binding site (marked with red box), homodimer interface (marked with green box) and catalytic residue (marked with orange box), and the AST pyridoxal-phosphate-binding site (IPR004838) referred to the sequence from 287 to 298 (Figure 4 and Figure 5C).

### 2.2. Phylogenetic Analysis

Three neighbor-joining (NJ) phylogenetic trees were constructed using the amino acid sequences of CfOpDHs, CfPEPCKs, and CfASTs and OpDH, PEPCK, and AST members of other species (Figure 6). As shown, the OpDH family was separated into two subfamilies, Porifera (ornithine cyclodeaminase (OCD)/mu-crystallin family) subfamilies (including OpDH of Porifera and OCD of other animals) and Lophotrochozoa (including OpDH of Cnidaria, Annelida, and Mollusca). Each subfamily is clustered according to the evolutionary status of the species. The phylogenetic tree of PEPCK was classified into two subfamilies, GTP-utilizing and ATP-utilizing. The GTP-utilizing subfamily was further divided into two classes, comprising a lower evolutionary status (including Nematoda and Mollusca) and a higher evolutionary status (including Chordata, Arthropoda, Cnidaria, and Mollusca). The two CfPEPCKs respectively located within two parts of the GTP-utilizing subfamily and firstly clustered with PEPCKs of other bivalves including *P. yessoensis*, *Pinctada imbricata*, and *Mytilus galloprovincialis.* The AST family is divided into three branches, representing three AST subfamilies, micro-organisms isoenzyme, the cytosolic isoenzyme, and the mitochondrial isoenzyme. Each AST subfamily is clustered according to the evolutionary status of the species. The two CfASTs respectively located within cytosolic and mitochondrial clades and firstly clustered with ASTs of other bivalves including *P. yessoensis*, *Ostrea edulis,* and *Crassostrea angulata*.

### 2.3. Spatiotemporal Expression

RNA-seq datasets for different developmental stages and adult tissues of *C. farreri* were used to detect the expression patterns of *OpDH*, *PEPCK*, and *AST* genes. As shown in Figure 7A, during the developmental process, *CfOpDH1/3*, *CfPEPCK1*, and *CfAST1/2* showed a high expression level. *CfOpDH2/5* and *CfPEPCK2* did not express or weakly expressed at early developmental stages but expressed during the middle and late developmental stages. Specifically, expression of *CfPEPCK1* gradually decreased from the zygote and maintained a comparatively low expression level from middle umbo to post umbo, while reaching a high level at eyespots larva and juvenile. The *CfAST2* was the highest expression member among genes, remaining at a continuously high expression level during developmental stages, and it reached a peak at the early umbo stage.

In all tested tissues, the expressions of *CfOpDH1/3*, *CfPEPCK1,* and *CfAST1/2* were ubiquitous, whereas *CfOpDH5* kept a quite low level (Figure 7B), which showed a similar expression pattern with developmental stages. *CfAST2* was the highest expression member among all genes followed by *CfAST1.* Generally, the genes *CfOpDH1/2/4*, *CfPEPCK1,* and *CfAST1/2* exhibited the highest expression levels in muscle and foot, followed by all genes except *CfOpDH4/5* in visceral ganglia and cerebral ganglia.

### 2.4. Expression Profiles of CfOpDHs, CfPEPCKs, and CfASTs in Response to Heat Stress

To examine the expression patterns of *CfOpDHs*, *CfPEPCKs*, and *CfASTs* in response to environmental stresses, RNA-seq datasets from *C. farreri* under heat stress were analyzed. In general, the expression of *CfOpDHs*, *CfPEPCKs*, and *CfASTs* changed apparently and responded frequently in mantle, gill, heart, hemolymph, digestive gland, and striated muscle (Figure 8). Specifically, in the examined tissues of the mantle, gill, heart, hemolymph, and digestive gland, the expression levels of *CfASTs* exhibited a general down-regulated pattern, while *CfPEPCKs* displayed a general up-regulated trend in all examined tissues. Additionally, *CfOpDHs* showed a general down-regulated expression profile, except *CfOpDH1/3*, which exhibited up-regulation in striated muscle at certain time points during heat stress.

In comparison to the control group, the expression of *CfPEPCK1* exhibited a significant increase at all time points in the mantle, gill, hemolymph, and striated muscle, and at some time points in the heart and digestive gland. Subsequently, the expression of *CfPEPCK2* also showed a significant rise at certain time points in the hemolymph, heart, and digestive gland (|log_2_FC| > 1 and *p* < 0.05). The expression of *CfOpDH1/3* increased significantly only in striated muscle at certain time points. Compared to gill and striated muscle, the expression levels of *CfOpDHs* and *CfASTs* did not exhibit statistically significant differences in the remaining four tissues.

Overall, the expression levels of *CfPEPCKs* remained up-regulated and *CfOpDHs* exhibited a tissue-specific expression pattern, whereas *CfASTs* showed a down-regulated trend under heat stress in *C. farreri*. Notably, *CfPEPCK1/2* exhibited more pronounced alternations compared to *CfOpDHs* and *CfASTs*, suggesting a primary role for *CfPEPCKs* in the response to heat stress.

## 3. Discussion

In recent years, there has been a notable increase in global marine environmental changes including rising temperatures and decreasing dissolved oxygen [14,39]. Anaerobic energy metabolism has been demonstrated to be essential for marine organisms to adapt to environmental disturbances [40,41,42,43]. In this study, we systematically identified the key enzymes involved in four anaerobic metabolism pathways in *C. farreri*. We then evaluated the expression of these enzymes across various development stages, different adult tissues, and under heat-stress conditions. Our findings present the initial genome-wide identification of the principal enzyme involved in four anaerobic metabolic pathways and offer valuable insights into the potential role of anaerobic metabolism in mollusks.

### 3.1. Glucose–Lactate Pathway

After extensive data analysis, the LDH was not identified in the *C. farreri* genome. This absence might explain the reason why lactate was not accumulated in sessile bivalves under anaerobic conditions [26,37]. Both LDH and OpDH pathways exhibited low efficiency yet high rates of energy production, thereby ensuring a constant flux of glycolysis and a steady supply of ATP by regulating the NADH/NAD+ ratio during hypoxia [23,34]. Increasing proline levels in *M. edulis* under short-term anaerobic stress could provide substrates for the glucose–opine pathway [13,17,35,36,44]. In addition, there was a study that proposed that the lactate pathway may have originated from opine pathways [13]. So, we suggested that the function of LDH in supplying rapid energy during anaerobic respiration may be achieved by OpDHs in *C. farreri*.

### 3.2. Glucose–Opine Pathway

Based on the ClustalW multiple sequence alignment results, it was evident that the OpDHs family members of marine invertebrates exhibited a widespread presence of conserved functional regions, such as the N-terminal glycine-rich GXGXXA/G motif [45]. This region constituted the Rossmann-fold structure essential for binding the co-factor NADH [46]. The presence of these conserved regions ensured the stable function of OpDHs throughout the evolutionary process of species. On account of the predicted results of OpDHs gene structure, it is obvious that they had a similar structural composition except CfOpDH5. CfOpDH5 was located on the same chromosome beside CfOpDH3, with only one coding sequence (CDS) region, and was significantly shorter in length compared to the remaining CfOpDHs. Multiple sequence alignment analyses revealed that CfOpDH5 closely matched CfOpDH3 at the majority of conserved sites. We suggested that CfOpDH5 may be a duplicate copy of CfOpDH3 and it was also coincident with the result of a phylogenetic tree. According to the phylogenetic tree analysis, the OpDHs of marine invertebrates can be classified into two clusters: sponge organisms and others (including mollusks, cnidarians, and annelids). Notably, the sponge OpDHs exhibited homology with proteins belonging to the OCD/mu-crystallin family, distinguishing them from other marine invertebrate taxa. The OCD/mu-crystallin protein family encompassed proteins with diverse functions and had homologs in all kingdoms of life [13,47]. According to previous studies, the distinction of different OpDH subfamilies was based on substrate specificity [48]. Studies detected the activity of StDH, AlaDH, TaDH, and OcDH in multiple tissues of *C. nobilis* [38], which was roughly consistent with the number of OpDHs identified in our study. However, according to phylogenetics, CfOpDHs did not cluster by their substrate specificity, which was consistent with previous studies [48]. One possible explanation is the similar and homologous sequences in the five CfOpDHs, for example, the 96% amino acid identity between beta-AlaDH and TaDH in *Cellana grata* [49], leading to their designation as CfOpDH1-5.

Different CfOpDHs exhibited disparate transcriptional levels and patterns during the developmental stages of *C. farreri* larvae. The expression of *CfOpDH1/3/4* in the multicellular stage may relate to the rapid proliferation of cells in the early stage of development, requiring energy in a short time and easily causing hypoxia [50]. The downregulation of *CfOpDH1/3/4* expression during the gastrula stage was evident, consistent with the finding in *Danio rerio* and *Montipora capitata*, suggesting a potential mechanism involving maternal RNA degradation and the zygotic genome activation [51,52,53]. Furthermore, the lower expression of *CfOpDHs* in gastrula might be attributed to the limited contribution of anaerobic metabolism to cell differentiation and organogenesis. A previous study in *Cynoglossus semibreves* suggested that energy generation predominantly occurs via the aerobic pathway during the gastrula stage [54]. In addition, the expression pattern of *OpDHs* in adult tissues was similar to the activity pattern observed in *C. nobilis*, with high levels of enzyme activity in the striated muscle, smooth muscle, and foot [38]. Studies showed that striated muscle contained more amino acids than other tissues, like arginine and glycine [55]. That may provide sufficient substrates for *CfOpDHs* and therefore tissues like muscle had more *CfOpDHs* to support the OpDH pathway. Moreover, studies showed that the high metabolic rate of *OpDH* could support for rapid movement of muscle; therefore, high *CfOpDH* expression could help meet the energy demand of muscle [34].

It has been suggested that the OpDHs pathway serves as a major pathway of anaerobic metabolism for energy demand in the early stages of hypoxia stress in marine invertebrates, confirmed by the level of octopine increased gradually after anaerobic stress [13,17,35,44,56]. Some gastropod mollusks, such as *C. toreuma*, also relied on the OpDHs pathway for anaerobic metabolism under heat stress. Precisely, the expression of AlaDH is up-regulated in the foot muscle at Arrhenius break temperature (ABT) (38 °C) to protect *C. toreuma* against heat stress during cardiac dysfunction [12]. Under 27 °C heat stress in *C. farreri*, the expressions of *CfOpDHs* were found to be up-regulated in some time points, specifically in the striated muscle among tested six tissues. This suggested that the specific function tissue of *CfOpDHs* was the muscle tissue. Studies showed that OpDH anaerobic metabolism not only happened during natural anaerobiosis but also occurred when energy demand was strongly increased. In *M. edulis*, the adductor muscle showed anaerobic end products formed such as trombone and octopine, under aerial exposure and first hours of reimmersion although PO_2_ was full restoration [57]. However, during the stress process, the transcriptional levels of *OpDHs* showed different degrees of down-regulation. That indicated that the anaerobic metabolism mechanism in response to heat stress varied among tissues or heat stress may adversely affect the activity of the enzyme with tissue-specific effects. To sum up, our results indicated that the OpDHs pathway is one of the important pathways for anaerobic metabolism under heat stress in the striated muscle of *C. farreri*.

### 3.3. Aspartate–Succinate Pathway

Another anaerobic metabolism pathway was the aspartate–succinate pathway. AST genes had similar structure and composition, on account of the results of the gene structure prediction. Based on multiple sequence alignment results, both AST family members had a pyridoxal-phosphate-binding site ranging from 287 to 300, with a lysine residue to covalent binds with coenzyme pyridoxal 5′-phosphate (PLP) [58,59]. The homology within AST subfamilies was significantly higher than the homology between different species. ASTs in marine invertebrates exhibited the same pattern as other species, clustering into two distinct branches: AST-C and AST-M [23]. In our study, two AST genes were identified with extremely high bootstrap values that cluster within the AST-C and AST-M branches of mollusks, respectively designated as CfAST1 and CfAST2.

In the aspartate–succinate pathway, AST catalyzed the conversion of aspartate to oxaloacetate, which was further converted via malate and fumarate to generate succinate, thereby maintaining the energy supply in organisms [17,27]. The ASTs identified in our study exhibited high transcriptional levels during most developmental stages of *C. farreri* larvae and in different tissues of adult individuals. That could be attributed to the pivotal role of AST in the metabolism of non-essential amino acids and the aspartate malate shuttle. AST facilitated the transamination process between aspartate and a-ketoglutarate, resulting in the production of glutamate and oxaloacetate, thereby influencing the regulation of carbon and nitrogen flux in amino acid metabolism [60]. Previous studies showed that the trochophore of bivalves exhibited robust locomotive capabilities [61]. The high expression of *CfAST2* from the trochophore stage might be the reason that the enhanced locomotor ability increased the frequency of locomotor hypoxia. Moreover, CfAST1 and CfAST2 might serve distinct functions, as evidenced by the consistent expression of *CfAST1* throughout *C. farreri* development, in contrast to the increasing expression of *CfAST2* from gastrula stage onwards, reaching exceptionally high levels in umbo larvae stages. AST was so common that it had high activity in almost all tissues, which was also reported in *Zalophus californianus* and *Phoca vitulina* [62]. The high expression of *CfASTs* in muscle, foot, and mantle consisted of the high enzyme activity observed in the same tissues of *C. nobilis* and *P. yessoensis* [38]. Notably, the muscle exhibited the highest expression and activity, indicating the importance of the AST pathway in energy metabolism specifically within muscle tissue.

According to previous research, during the early stages of anaerobic metabolism under hypoxia stress in marine invertebrates, AST transaminated to produce alanine following glycolysis, which was consistent with the increase in alanine content [21,36]. During both early and late stages of heat stress at 27 °C in *C. farreri*, the transcriptional levels of *ASTs* were significantly decreased in both gill and striated muscle tissues. We speculated that the gill was directly exposed to the heated sea water, which impacted the activity and expression of *CfASTs*. The striated muscle contained OpDH3 to regulate amino acid metabolism, suggesting a modest reduction in activity could be beneficial for conserving energy. Furthermore, the expression of *CfASTs* remained relatively stable under heat stress, suggesting that the high expression levels were maintained and the essential functions were preserved.

### 3.4. Glucose–Succinate Pathway

The last anaerobic metabolism pathway was the glucose–succinate pathway. On account of the results of PEPCK gene structure prediction, it was obvious that they had a similar structural composition. Based on the multiple sequence alignment results, the PEPCK family members shared a perfectly conserved site in the central part of PEPCK, with an essential cysteine residue in the center proposed to be implicated in the catalytic mechanism [63]. As shown, PEPCKs in marine invertebrates could be classified into two clusters: PEPCK-C and PEPCK. The PEPCK branch was the ancestral cluster and is located between the ATP cluster and the GTP cluster evolution node, while the PEPCK-C branch exhibited higher homology with its counterparts in higher vertebrates. We speculated that the differentiation of PEPCKs in marine invertebrates occurred gradually and was distinct from the co-evolution of AST-C and AST-M. Some PEPCKs retained the original structure of PEPCK-GTP, like PEPCK in *Caenorhabditis elegans*; while the others evolved into subfamily structures (PEPCK-C and PEPCK-M) as the result of long-term localization in the cytoplasm and mitochondria, which made them specialize. This kind of evolution progressed variously in different marine invertebrates, from no-evolved (PEPCK) in *C. elegans*, to only one evolved (PEPCK and PEPCK-C) in mollusks and to both evolved (PEPCK-M and PEPCK-C) in cnidaria. It eventually led to the formation of the PEPCK-C and PEPCK-M subfamilies in vertebrates, with the homology between PEPCK subfamilies significantly higher than between species [64]. CfPEPCKs are respectively clustered in the PEPCK-C and PEPCK branches of mollusks, designated as CfPEPCK1 (cytosolic isoform) and CfPEPCK2 (ancestral isoform).

The expression of *CfPEPCK1* decreased gradually until the eyespots larva stage, whereas *CfPEPCK2* exhibited a gradual upregulation, particularly after D-stage veliger. Previous studies have indicated that eyespots played a role in response to light or shadows of mollusks, and neurogenesis initiated at the early veliger stage [65,66]. It is hypothesized that the elevated expression of *CfPEPCKs* during these developmental stages may be necessary to support energy requirements for light reactions and ganglia development. Furthermore, high transcriptional levels of *CfPEPCK1* in most tissues of adult *C. farreri* were distinct from the low PEPCK activities detected in *C. nobilis* [38]. However, studies showed that changes in concentrations of fructose bisphosphate, alanine, and ITP controlled the activities of PEPCK in the *Metridium senile* and *O. edulis* muscles [67]. Therefore, we proposed that *PEPCKs* maintained a certain level of expression in scallops and had separate activity levels adjusted to meet physiological demands. Studies showed that the striated muscle was an important motor and feeding organ and the gill was a vital respiratory organ whose changes under stress had crucial effects on mollusks [68,69]. Therefore, the high expression levels of *CfPEPCK1* in striated muscle and gill suggested its potential significance in the regulation of movement metabolism and stress response. Conversely, the elevated expression of both *CfPEPCK1* and *CfPEPCK2* in cerebral ganglia and visceral ganglia indicated its potential role as a crucial regulator in metabolic adjustment and control, given the involvement of ganglia in the coordination of bodily functions and movement.

Studies showed that the mitochondrial form of *PEPCK* in vertebrates was widely expressed, which could be induced by diverse stress, and two forms of PEPCK played different roles in metabolism [70,71,72]. According to existing studies, during prolonged anaerobic stress in marine invertebrates, in the glucose–succinate pathway, PEPCK catalyzed the conversion of phosphoenolpyruvate to oxaloacetate, eventually leading to succinate, as well as volatile fatty acids (acetate and propionate), to maintain the organism’s energy supply [17,27,30]. According to the expression profiles of *CfPEPCKs*, it was observed that *CfPEPCK2* significantly upregulated in the heart, hemolymph, and digestive glands at early and middle time points, which suggested a potential regulatory role under heat stress. Significantly, *CfPEPCK1* showed a pronounced upregulation in all examined tissues throughout the 27 °C heat-stress process. This observation strongly implied the pivotal involvement of PEPCK in the cellular response to heat stress, further suggesting the glucose–succinate pathway, catalyzed by PEPCK, served as the predominant anaerobic metabolism pathway during such stress. Previous research demonstrated a linear increase in the accumulation of succinate and propionate in *M. edulis* as the duration of anaerobic stress increases, which were intermediate metabolites or end products of the glucose–succinate pathway [36,73]. The glucose–succinate pathway was also proved to play a vital role when *P. yessoensis* faced drying stress [30].

Summing up the above, we suggested that the glucose–succinate pathway, catalyzed by CfPEPCK1, is the main pathway for anaerobic metabolism under heat stress in *C. farreri*. An anaerobic metabolic network was created according to this research, helping provide a more complete understanding of the anaerobic metabolism under heat stress in *C. farreri* (Figure 9).

## 4. Materials and Methods

### 4.1. Genome-Wide Identification and Sequence Analysis of LDH, OpDH, PEPCK, and AST Genes in C. farreri

The whole genome and transcriptome databases of *C. farreri* (PRJNA259405 and SRA027310) were used to search with the representative LDH, OpDH, PEPCK, and AST protein sequences of other species, retrieved from NCBI (https://www.ncbi.nlm.nih.gov/guide/proteins/ (accessed on 3 October 2022)) (Appendix A) [74]. ORF (open reading frame) finder was devoted to predicting their gene structure (http://www.ncbi.nlm.nih.gov/orffinder/ (accessed on 26 March 2023)). The amino acid sequences were confirmed by BLASTP against the NCBI non-redundant protein sequence database. InterPro (Classification of protein families) (https://www.ebi.ac.uk/interpro/ (accessed on 26 March 2023)) predicted the conserved domains. The putative isoelectric point (PI) and molecular weight were predicted by the ExPASy-ProtParam (https://web.expasy.org/protparam/ (accessed on 27 January 2024)). The secondary and tertiary structures were predicted by the software Geneious 4.8.4 (https://www.geneious.com/ (accessed on 26 March 2023)) and Phyre^2^ (Protein Homology/analogy Recognition Engine V 2.0) (http://www.sbg.bio.ic.ac.uk/phyre2/html/page.cgi?id=index (accessed on 26 March 2023)).

### 4.2. Phylogenetic Analysis

The identified OpDH, PEPCK, and AST protein sequences from *C. farreri* and other organisms were performed for multiple protein sequence alignments by the ClustalW program in the software MEGA 11 [75]. The phylogenetic tree of each protein family was constructed utilizing the software MEGA 11 with the neighbor-joining method (Appendix A) [75]. Bootstrapping with 10,000 replications was used to evaluate the phylogenetic tree.

### 4.3. Spatiotemporal Expression of OpDH, PEPCK, and AST Genes in C. farreri

For expression analysis, the RPKM (reads per kilo per million reads) values of *CfOpDHs*, *CfPEPCKs*, and *CfASTs* were retrieved from RNA-seq datasets of *C. farreri* [74]. The RPKM values, including various developmental stages (zygote, multicell, blastula, gastrula, trochophore, D-shaped larvae, early umbo, middle umbo, post umbo, eyespots larvae, and juvenile) and adult tissues (striated muscle, smooth muscle, foot, eye, mantle, gill, digestive gland, hemolymph, kidney, male gonad, female gonad, visceral ganglia, and cerebral ganglia) (Appendix A), were Log_2_(x + 1) transformed and subsequently used to draw a heat map by OmicStudio tools (https://www.omicstudio.cn (accessed on 30 March 2023)).

### 4.4. Expression Analysis of CfOpDHs, CfPEPCKs, and CfASTs under Heat Stress

The transcriptome databases of *C. farreri* in response to heat stress were independently constructed by our laboratory [76]. Specifically, the thermal stimulation was set as 27 °C, which is a temperature closely resembling the maximum recorded in the sea areas where scallops are typically cultured. The temperature of the seawater was regulated using thermostatic heaters (BOYU, HT2200, China). In addition, a control condition of 20 °C was employed, which was consistent with the temperature of the sampled location. Transcriptome databases at eight time points (3 h, 6 h, 12 h, 24 h, 3 d, 6 d, 15 d, and 30 d) of six tissues, including mantle, gill, heart, hemolymph, striated muscle, and digestive gland, were used to reveal the expression levels of *CfOpDHs*, *CfPEPCKs*, and *CfASTs* under heat stress. Expression of all those genes was calculated with the form of count (Gene Expression Counts) and log_2_FC (Fold change) was calculated between each heat stress time point and control group (Appendix A). Differentially expressed genes were identified using an edgeR package with a statistically significant cutoff of |log_2_FC| > 1 and *p*-value < 0.05 (Appendix A). The log_2_FC values were used to perform a heat map by OmicStudio tools (https://www.omicstudio.cn (accessed on 30 March 2023)).

## 5. Conclusions

In summary, we successfully identified five CfOpDHs, two CfASTs, and two CfPEPCKs from the *C. farreri* genome. Different subtypes of *CfOpDH1-5*, *CfAST1/2*, and *CfPEPCK1/2* showed varying expression levels during larval development and in adult tissues, indicating their involvement in the energy supply for growth, development, and physiological processes in *C. farreri*. Under heat stress, the expression of *CfASTs* showed an overall downregulation trend in all six tissues, while the expression of *CfPEPCK1* in all test tissues and *CfOpDH1/3* in striated muscle exhibited a remarkable upregulation trend. Those distinct expression patterns suggest the mechanism of adaptation to temperature changes in *C. farreri* occurs through the three anaerobic metabolism pathways synergistically. The glucose–succinate pathway, catalyzed by CfPEPCK1, is the predominant pathway for anaerobic metabolism under heat stress in all tissues, with *CfOpDH1/3* catalyzing the glucose–opine pathway in striated muscle as supplementary. The findings of this study provide insights into the main pathways of anaerobic metabolism in scallops and shed light on the evolution of OpDH, AST, and PEPCK in marine invertebrates. Further research on the functions of *CfPEPCKs* will provide a more detailed understanding of the energetic regulation of bivalve organisms under environmental stress.

## Figures and Tables

**Figure 1 ijms-25-04741-f001:**
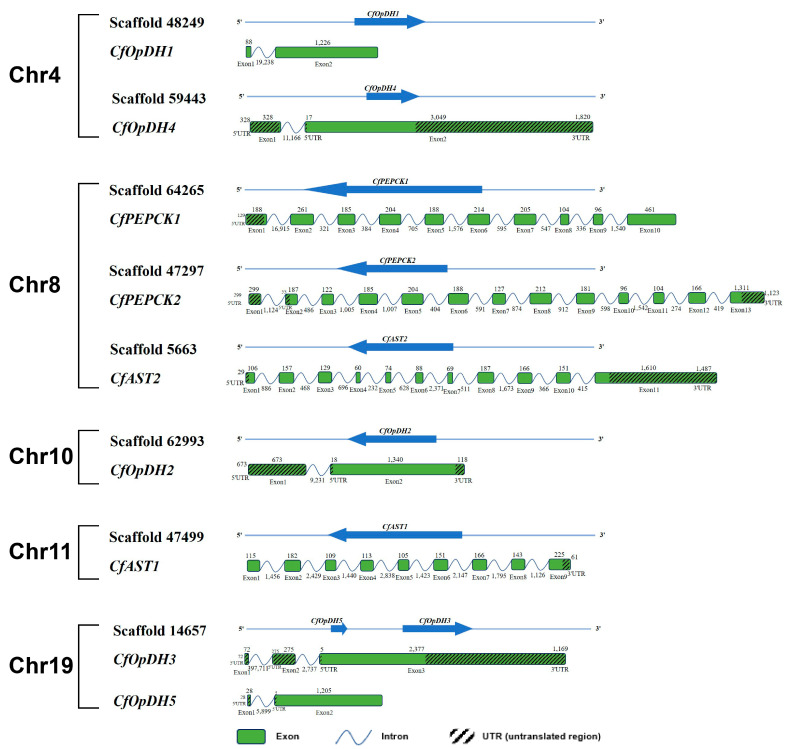
Genomic location and gene structure of CfOpDHs, CfPEPCKs, and CfASTs genes. The blue arrows on the lines indicate the code direction of genes inserted in scaffolds. The green boxes indicate the exons. The polylines indicate the intron. The diagonal boxes indicate the 5′ UTRs and 3′ UTRs. The 3′ UTR, 5′ UTRs, and exons are shown according to their lengths in the DNA sequence. The numbers above the boxes indicate the length of 3′ UTR, 5′ UTR, and exon, the numbers below the polylines indicate the length of the intron.

**Figure 2 ijms-25-04741-f002:**
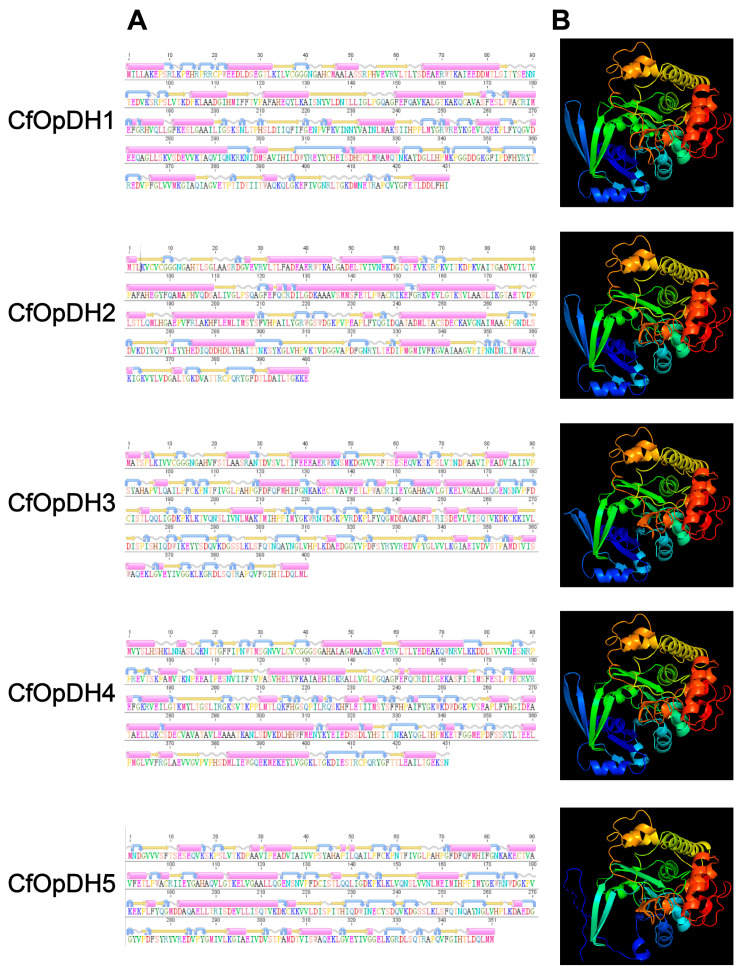
Predicted secondary structures (**A**) and tertiary structures (**B**) of CfOpDHs. The pink cylinders stand for alpha helixes, the orange straight arrows represent beta strands, the grey wavy lines represent coils, and the blue curved arrows represent turns.

**Figure 3 ijms-25-04741-f003:**
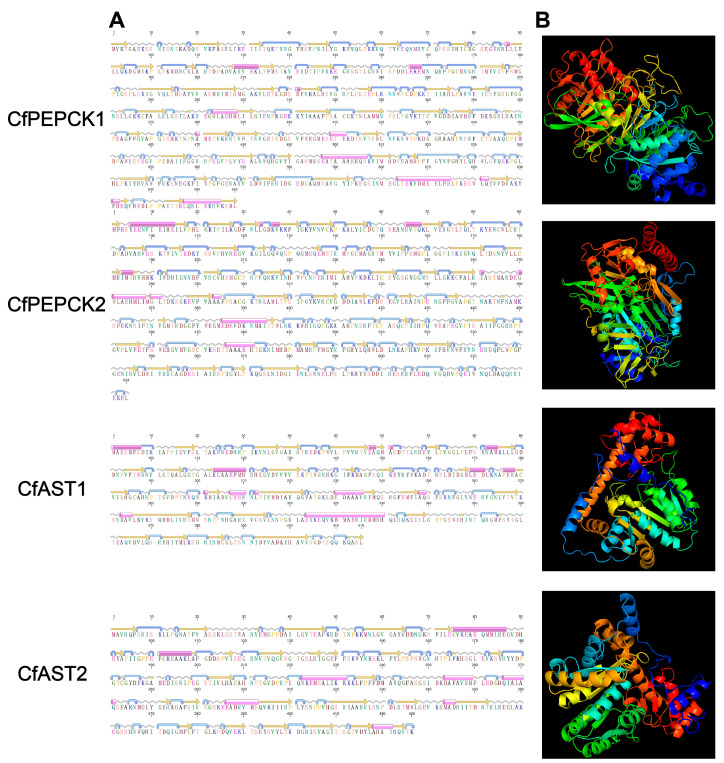
Predicted secondary structures (**A**) and tertiary structures (**B**) of CfPEPCKs and CfASTs. The pink cylinders stand for alpha helixes, the orange straight arrows represent beta strands, the grey wavy lines represent coils, and the blue curved arrows represent turns.

**Figure 4 ijms-25-04741-f004:**
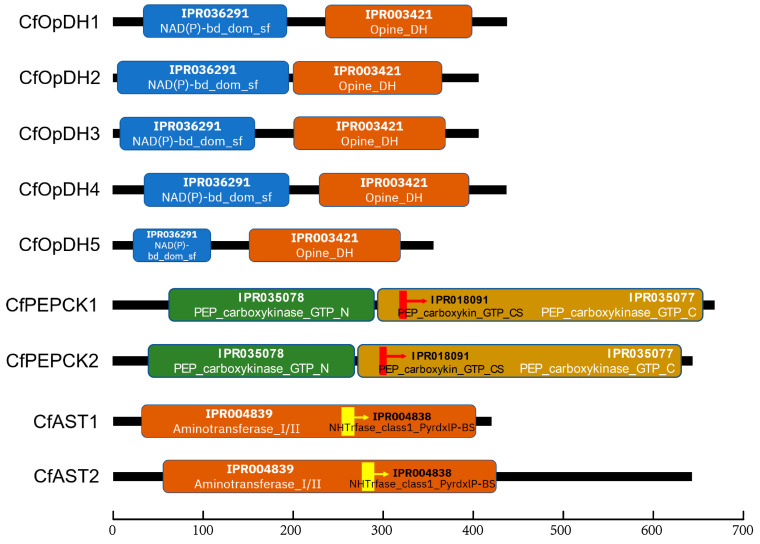
Predicted protein domain architecture of CfOpDHs, CfPEPCKs, and CfASTs by InterPro. A NAD(P)-binding domain superfamily (blue box) and an Opine dehydrogenase (chocolate box) are respectively shown in CfOpDHs. A phosphoenolpyruvate carboxykinase GTP-utilizing N-terminal (green box), a phosphoenolpyruvate carboxykinase GTP-utilizing conserved site (red stick with an arrow), and a phosphoenolpyruvate carboxykinase C-terminal P-loop domain (gold enrod box) are respectively shown in CfPEPCKs. An Aminotransferase classⅠ/classⅡ (chocolate box) and an Aminotransferases classⅠ pyridoxal-phosphate-binding site (yellow stick with an arrow) are respectively shown in CfASTs.

**Figure 5 ijms-25-04741-f005:**
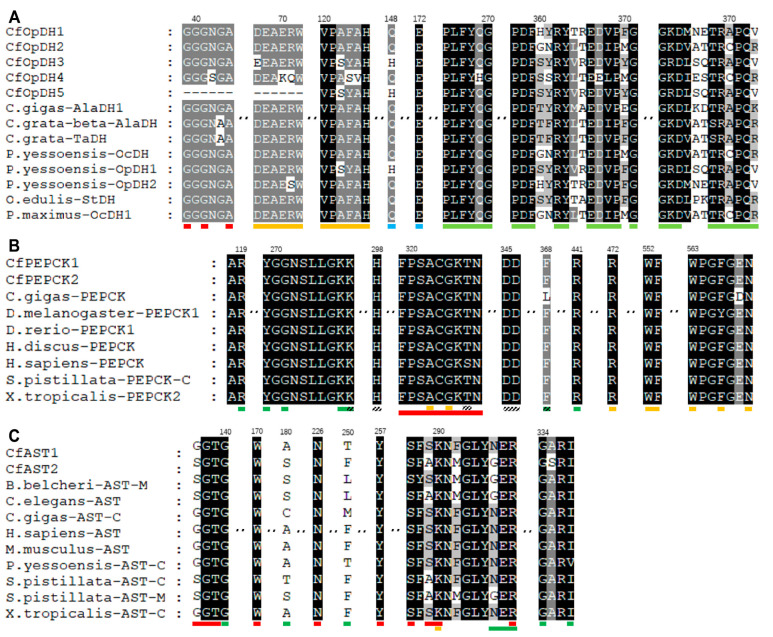
Amino acid sequences alignment of conserved domains for CfOpDHs, CfPEPCKs, and CfASTs, and other OpDHs, PEPCKs, and ASTs. Black background, grey background with white letters, and black letters respectively represent conserved value 100%, 80% and below 80%. (**A**) In OpDHs, the red boxes indicate the conserved Rossmann-fold motifs (GXGXXA/G) of NAD(P)-binding domain, the orange boxes indicate the n-terminal conserved sequences (DEAERW and VPAFAH), the blue boxes indicate the binding site of L-Arginine and pyruvate and the green boxes indicate the conserved sequences related to substrate binding (PLFYH/QG, PDFXXRYXXEDI/VPXG, and GKDXXXTRA/CPQR). (**B**) In PEPCKs, the green boxes indicate the substrate-binding site, the diagonal boxes indicate the metal-binding site, the orange boxes indicate the GTP-binding site, and the red boxes indicate the conserved site (signature). (**C**) In ASTs, the red boxes indicate the pyridoxal 5′-phosphate binding site, the green boxes indicate the homodimer interface, and the orange box indicates the catalytic residue.

**Figure 6 ijms-25-04741-f006:**
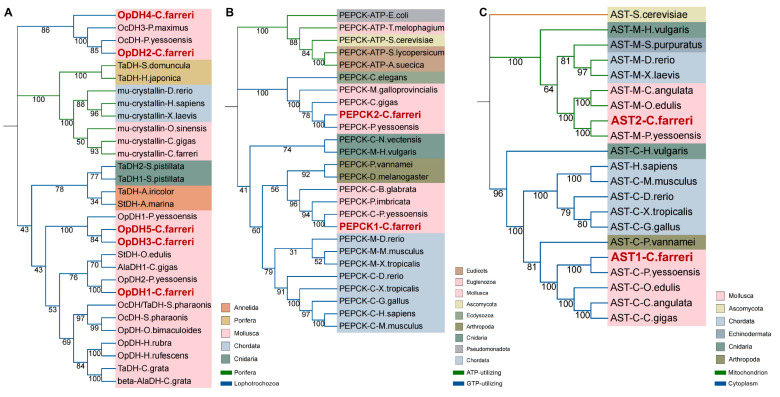
Three phylogenetic trees were constructed based on the protein sequences of CfOpDHs, CfPEPCKs, CfASTs, and other species. Numbers at tree nodes indicate the bootstrap values from 10,000 replicates. The red name indicates CfOpDHs, CfPEPCKs, and CfASTs. (**A**) In the tree of OpDHs, the green branches are Porifera subfamilies and the blue branches are Lophotrochozoa subfamilies. (**B**) In the tree of PEPCKs, the blue branches are GTP-utilizing subfamilies and the green branches are ATP-utilizing subfamilies. (**C**) In the tree of ASTs, the green branches are mitochondrion subfamilies and the blue branches are cytoplasm subfamilies. The colors of the rings represent different phyla.

**Figure 7 ijms-25-04741-f007:**
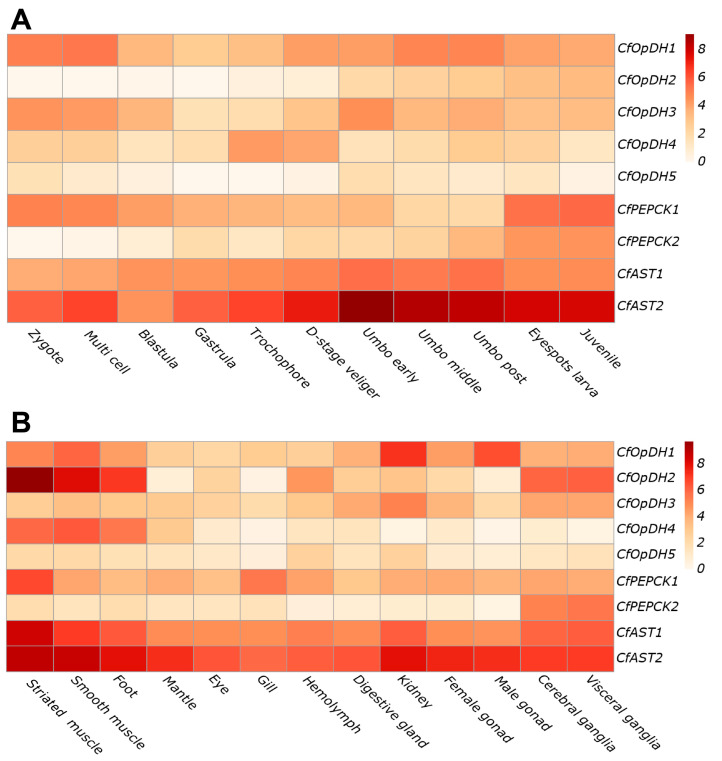
Expression analysis heat maps of *CfOpDHs*, *CfPEPCKs,* and *CfASTs* during different developmental stages (**A**) and in adult tissues (**B**) of *C. farreri* based on the log_2_RPKM.

**Figure 8 ijms-25-04741-f008:**
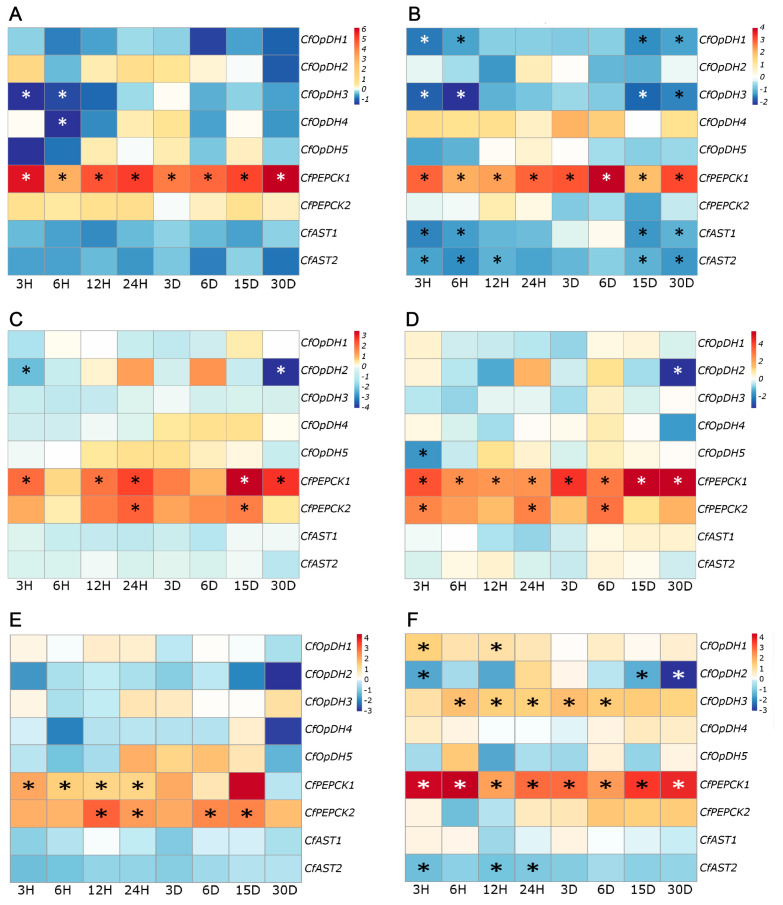
Expression of *CfOpDHs*, *CfPEPCKs*, and *CfASTs* in the mantle (**A**), gill (**B**), heart (**C**), hemolymph (**D**), digestive gland (**E**), and striated muscle (**F**) of *C. farreri* under heat stress at different time points based on the log_2_FC value. The expression of *CfOpDHs*, *CfPEPCKs*, and *CfASTs* at 0 h was used as the control. Values marked with asterisks indicate significant differences from the control expression (* |log_2_FC| > 1 and *p* < 0.05).

**Figure 9 ijms-25-04741-f009:**
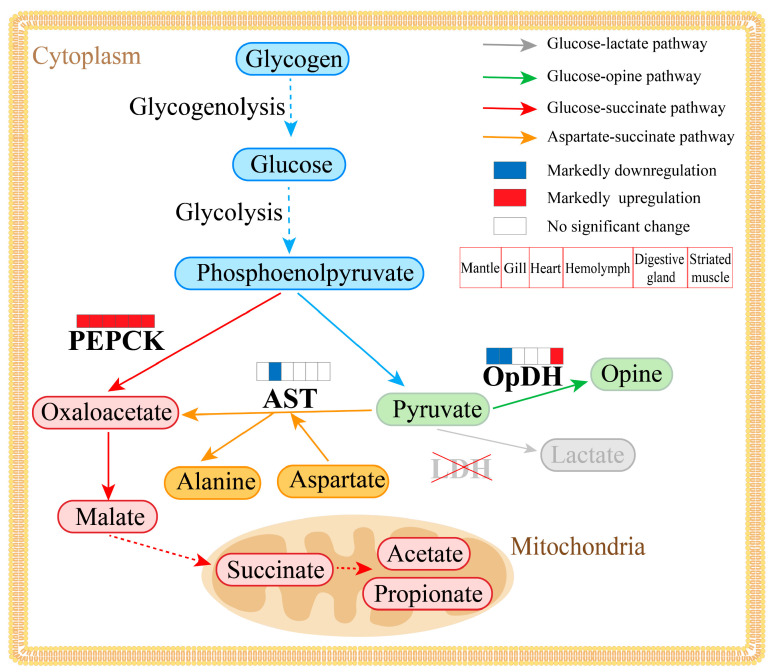
Schematic overviews of main anaerobic metabolic pathways, involved in different tissues (mantle, gill, heart, hemolymph, digestive gland, and striated muscle) of *C*. *farreri*. LDH, lactate dehydrogenase; OpDH, opine dehydrogenase; PEPCK, phosphoenolpyruvate carboxykinase; AST, aspartate aminotransferase. Arrows and oval boxes represent the key enzyme and intermediate or end products of pathways, respectively. Gray, green, red, orange and blue respectively represents the glucose-lactate, glucose-opine, glucose-succinate, aspartate-succinate pathway and the common process before anaerobic metabolism. The squares above the key enzymes respectively represent their expression trends in mantle, gill, heart, hemolymph, digestive gland, and striated muscle under heat stress. Blue, red and white respectively represents markedly downregulation, markedly upregulation and no significant change.

**Table 1 ijms-25-04741-t001:** The sequence characteristics of identified genes.

Name	CfOpDH1	CfOpDH2	CfOpDH3	CfOpDH4	CfOpDH5	CfPEPCK1	CfPEPCK2	CfAST1	CfAST2
Gene ID	scaffold48249.6	scaffold62993.3	scaffold14657.35	scaffold59443.2	scaffold14657.34	scaffold64265.9	scaffold47297.11	scaffold47499.10	scaffold5663.76
Total length (bp)	20,552	11,244	45,172	14,543	7132	25,025	12,618	15,963	11,043
ORF length (bp)	1314	1203	1203	1212	1200	1977	1905	1248	1281
5′ UTR length (bp)	---	692	352	345	33	129	354	127	29
3′ UTR length (bp)	---	118	1169	1820	---	---	1123	61	1487
Amino acids length	437	400	400	403	399	658	634	415	426
Weight (kDa)	48.49	43.44	44.06	48.17	38.91	73.47	71.65	46.53	47.18
Theoretical pI	5.99	5.25	5.75	6.19	5.22	7.56	8.20	6.61	8.55
Number of exons	2	2	3	2	2	10	13	9	11
Number of introns	1	1	2	1	1	9	12	8	10
Number of alpha helixes	23	20	23	26	21	12	11	13	8
Number of beta strands	20	23	34	22	28	50	53	32	31
Number of coils	32	25	34	34	30	70	70	42	38
Number of turns	31	24	38	32	33	66	78	42	44

## Data Availability

Data is contained within the article and Appendix A.

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
