# Peer review of "The Glucose–Succinate Pathway: A Crucial Anaerobic Metabolic Pathway in the Scallop Chlamys farreri Experiencing Heat Stress"

_ijms, 2024, doi:10.3390/ijms25094741_

Round 1
Reviewer 1 Report
Comments and Suggestions for Authors
This paper presents the findings from a biochemistry study to investigate anaerobic metabolism in shellfish, specifically the glucose-succinate pathway in the Scallop Chlamys farreri during heat stress. The authors provided convincing results studying key enzymes of four anaerobic metabolic pathways: opine dehydrogenase genes (CfOpDHs), aspartate aminotransferase genes (CfASTs), phosphoenolpyruvate carboxykinase genes (CfPEPCKs) and lactate dehydrogenase (LDH). In accordance with their results LDH was absent in the genome of scallops, as well as under heat stress, the expression of CfASTs in scallops exhibited a general downregulation trend in the six tissues examined. I believe there is much merit to the data presented in this paper and the findings are very important. The manuscript is well written and formally presented. I believe it contributes to the development in this area of knowledge. I recommend its publication in this renowned scientific journal.
Reviewer 2 Report
Comments and Suggestions for Authors
The study of Bao et al examine the effect of heat stress on Anaerobic Metabolic Pathway in the Scallop Chlamys farreri. Whereas, elevated seawater temperatures hinder shellfish growth and can lead to mass mortality events. Understanding anaerobic metabolism pathways, crucial for survival under heat stress, remains a research focus, especially regarding enzyme activity and end products in marine invertebrates. A study on Chlamys farreri aims to identify and analyze key enzymes involved in anaerobic metabolism, providing insights into shellfish resilience under changing environmental conditions.
This study provides a comprehensive and systematic analysis of the key enzymes involved in anaerobic metabolism pathways, which holds significant importance in understanding the mechanism of energy supply in shellfish.
This work is interesting and all section well prepared. Just few comments including:
- Shortened the introduction.
- Only materials and methods section need some improvement related to the study animals including the age, weight, rearing condition, if it cultured in lab. or farm the type of feed and its composition. or caught from the natural water body (in this case, describe the handling conditions).
- How heat stress was induced and confirmed.
- The sampling procedures.
The resolution of figure 1-5 is very low, it could improve to be readable.
The results and discussion are well described and discussed.
The conclusion is prepared in line with the obtained results.
Reviewer 3 Report
Comments and Suggestions for Authors
The article titled "The Glucose-Succinate Pathway: A Crucial Anaerobic Metabolic Pathway in the Scallop Chlamys farreri Experiencing Heat Stress" presents a comprehensive investigation into the anaerobic metabolic pathways of the scallop Chlamys farreri under heat stress conditions. The study identifies key enzymes involved in anaerobic metabolism and explores their expression patterns during development, in various tissues, and under heat stress.
The introduction section sets a solid foundation for the study by contextualizing the significance of scallops in China's mariculture industry as the most important issue and then highlighting the environmental challenges posed by rising sea temperatures. However, it is a bit far from the title and the aim of the article. From that point, the introduction could be revised to more prominently emphasize the unique insights gained from the systematic identification and evolutionary analysis of key enzymes in Chlamys farreri and their expression patterns under heat stress.
The quality of English in the article is generally good, however, there are a few instances of awkward phrasing and minor grammatical errors that could be improved for smoother readability.
L.76 oxygen solubility in seawater instead of … oxygen solubility of seawater
L327 key enzymes instead of … key enzyme
Comments on the Quality of English LanguageThe quality of English in the article is generally good, however, there are a few instances of awkward phrasing and minor grammatical errors that could be improved for smoother readability.
Reviewer 4 Report
Comments and Suggestions for Authors
Dear authors, I find your study intriguing; however, I have some comments for improvement:
Terminology – arrange the termins ,,shellfish'', ,,mollusks'', ,,bivalves'' used throughout the text.
Please write the section Material and methods in detail:
- the Scallops and housing
- please write about the experimental procedures
- where did you sample the scallops?
- how did you stimulate gonad production?
- how did you stimulate the thermal stress?
- did you feed the scallops during experiments?
- how many times did you repeat the experiment?
- write about cell and tissue sample collection.
- write about RNA isolation.
- which primers did you use?
Round 2
Reviewer 4 Report
Comments and Suggestions for Authors
Dear authors, thank you for all the answers and improvements. I recommend the Manuscript for publication.